**Data Availability Statement:** We have conducted our study mainly on the "MosMedData", which is freely available for the purpose of research. MosMedData: Chest CT Scans with COVID-19

# AI-driven quantification of ground glass opacities in lungs of COVID-19 patients using 3D computed tomography imaging

**Monjoy Saha**[1]*, **Sagar B. Amin**[2], **Ashish Sharma**[1], **T. K. Satish Kumar**[3], **Rajiv K. Kalia**[3,4,5,6]

1 Department of Biomedical Informatics, Emory University School of Medicine, Atlanta, GA, United States of America, 2 Department of Radiology and Imaging Sciences, Emory University School of Medicine, Atlanta, GA, United States of America, 3 Department of Computer Science, University of Southern California, Los Angeles, CA, United States of America, 4 Collaboratory for Advanced Computing and Simulations, University of Southern California, Los Angeles, CA, United States of America, 5 Department of Physics & Astronomy, University of Southern California, Los Angeles, CA, United States of America, 6 Department of Chemical Engineering and Materials Science, University of Southern California, Los Angeles, CA, United States of America

* monjoybme@gmail.com

## Abstract

### Objectives

Ground-glass opacity (GGO)—a hazy, gray appearing density on computed tomography (CT) of lungs—is one of the hallmark features of SARS-CoV-2 in COVID-19 patients. This AI-driven study is focused on segmentation, morphology, and distribution patterns of GGOs.

### Method

We use an AI-driven unsupervised machine learning approach called PointNet++ to detect and quantify GGOs in CT scans of COVID-19 patients and to assess the severity of the disease. We have conducted our study on the "MosMedData", which contains CT lung scans of 1110 patients with or without COVID-19 infections. We quantify the morphologies of GGOs using Minkowski tensors and compute the abnormality score of individual regions of segmented lung and GGOs.

### Results

PointNet++ detects GGOs with the highest evaluation accuracy (98%), average class accuracy (95%), and intersection over union (92%) using only a fraction of 3D data. On average, the shapes of GGOs in the COVID-19 datasets deviate from sphericity by 15% and anisotropies in GGOs are dominated by dipole and hexapole components. These anisotropies may help to quantitatively delineate GGOs of COVID-19 from other lung diseases.

Related Findings; https://mosmed.ai/datasets/covid19_1110.

**Funding:** Ashish Sharma would like to acknowledge support from the National Cancer Institute U24CA215109. The content of this publication does not necessarily reflect the views or policies of the Department of Health and Human Services, nor does mention of trade names, commercial products, or organizations imply endorsement by the U.S. Government. T. K. Satish Kumar and Rajiv K. Kalia (RKK) would like to acknowledge the support of Zumberge Research and Innovation Fund at the University of Southern California. RKK would like to thank Sarah Kalia for helpful discussions. The funders had no role in study design, data collection and analysis, decision to publish, or preparation of the manuscript.

**Competing interests:** The authors declare that there is no conflict of interest.

## Conclusion

The PointNet++ and the Minkowski tensor based morphological approach together with abnormality analysis will provide radiologists and clinicians with a valuable set of tools when interpreting CT lung scans of COVID-19 patients. Implementation would be particularly useful in countries severely devastated by COVID-19 such as India, where the number of cases has outstripped available resources creating delays or even breakdowns in patient care. This AI-driven approach synthesizes both the unique GGO distribution pattern and severity of the disease to allow for more efficient diagnosis, triaging and conservation of limited resources.

## §1 Introduction

The COVID-19 pandemic has overwhelmed the world. It has infected over 342 million people and killed over 5.5 million people at the time of writing this paper (source: https://coronavirus.jhu.edu/map.html). The world economy has experienced shutdowns in response to the health crisis. This highly contagious disease, caused by severe acute respiratory syndrome coronavirus 2 (SARS-CoV-2), mainly affects the human respiratory system [1]. The clinical symptoms of the disease range from asymptomatic infection to coughing, body aches, fatigue, fever, shortness of breath, vomiting, loss of smell and taste, and, in severe cases, respiratory failure and death [2]. COVID-19 has a higher incidence among males than females. Older people ($> 60$ years) with underlying conditions such as obesity, hypertension, deficient immune system, and heart disease are at increased risk for developing severe illness [3].

Early detection and quantification of COVID-19 are direly needed to reduce the spread and mitigation of the disease. Two commonly used diagnostic and assessment tests are the reverse transcriptase-polymerase chain reaction (RT-PCR) and imaging testing such as chest radiograph and computed tomography (CT) [4]. The RT-PCR testing has been below par so far for two reasons: long processing time and poor sensitivity resulting in false negatives. The RT-PCR results depend critically on the rate of viral expression at the time of specimen collection.

The multinational Fleischner society emphasizes the importance of chest CT and chest radiographs in detecting and managing COVID-19 based on patient symptoms, pre-test probability, risk factors, and available resources [5]. Typical imaging findings of COVID-19 include ground glass opacities (GGO) in a bilateral and peripheral distribution or multifocal round GGOs [6]. This can occur with or without interlobular septal thickening and intralobular lines, commonly referred to as a crazy-paving pattern [7]. Features of organizing pneumonia can also be seen later in the disease process. The disease manifestation in CT scans depends on when the scans are taken. Chest CTs may look normal at the onset of the disease and GGO-related features tend to peak 9–13 days after the onset of the disease. CT can be used to determine the severity of the disease, which can help triage resources from patients with mild symptoms to those with severe symptoms. CT is also helpful in identifying asymptomatic patients in communities with widespread transmission of COVID-19.

In countries overrun by the pandemic, rapid and accurate detection of GGOs to diagnose and triage patients with COVID-19 can help improve efficiency and conserve resources. Since the volume of exams can be overwhelming for radiologists, several computational groups around the world have resorted to using machine learning (ML) to detect GGOs on CT exams.

As described in the next section, most of the work is based on supervised deep learning techniques in segmenting tumors in medical images. Our approach to GGO analysis is different and has four distinguishing features:

- We combine an unsupervised computer vision approach with convex hull and convex points algorithms to segment and preserve the actual structure of the lung.

- To the best of our knowledge, we are the first group to use PointNet++ architecture for 3D visualization, segmentation, classification, and pattern analysis of GGOs.

- We make abnormality predictions using a deep network and Cox proportional-hazards model using lung CT images of COVID-19 patients.

- We quantify the shapes and sizes of GGOs using Minkowski tensors to understand the morphological variations of GGOs within the COVID-19 cohort.

The paper is organized as follows: In the next section, we briefly review the literature on the segmentation of GGOs and lungs on chest CT exams of COVID-19 patients using deep learning approaches. In section 3, we describe the data resource and our approach to segmentation of lungs and GGOs. In section 4, we present results on detection, segmentation, abnormality predictions, and shape and size analyses of GGOs. Section 5 contains conclusions.

## §2 Related work

Rapid and accurate segmentation of CT scans to assess the severity of the disease in COVID-19 patients presents several challenges for ML methods: blurring due to low contrast in intensities of infected and normal tissues, significant variations in infection features, and insufficient training data. In CT scans of COVID-19 patients, GGOs appear at the periphery of both lungs. The pixel intensities of GGOs and soft tissue are nearly the same, which makes it difficult for non-clinicians to delineate GGOs and soft tissue regions. Various groups have proposed conventional and advanced ML approaches to segment the lung regions. In [8], a Convolutional Neural Network (CNN) Mask R-CNN approach is combined with Bayes, Support Vectors Machine (SVM), $K$-means, and Gaussian Mixture Models (GMMs) to segment lungs in CT scans. This combined supervised and unsupervised ML approach has an accuracy of 97.68 ± 3.42% and an average runtime of 11.2 seconds for lung segmentation. The results are not surprising because the images are CT scans of healthy patients and therefore not too complicated for segmentation. In [9], the authors utilized a Generative Adversarial Network (GAN) for lung segmentation in CT images. Note that GAN is commonly used for data augmentation and synthesis. The images presented in [9] are only from healthy lung scans. There are no signs of infections on those lung scans. Infected lung scans contain diffused regions, which are much harder to segment by ML algorithms. In [10], a random walker algorithm is used for lung segmentation. In this approach, a gray value is randomly sampled, and the GMM is used to compute the clustering probability. Subsequently, a random walker map is constructed by combining the clustering result with the spatial distance to compute new edge weights. The authors of [10] tested their methodology on 65 CT scans and reported 97% to 99% accuracy for two different datasets. A CNN-based architecture was proposed for lung segmentation in [11]. The authors reported a 98.5% dice coefficient value for this study. Recently, the Inf-Net architecture consisting of multiple convolutional, up-sampling, and down-sampling layers was developed for lung segmentation and applied to datasets of COVID-19 patients [12]. The results for GGO segmentation do not match very well with the ground truth, which indicates that this approach requires further improvements and more training data.

Table 1 summarizes ML studies of lung segmentation in CT scans of healthy and COVID-19 patients.

## §3 Materials and methods

We have conducted our study mainly on the "MosMedData: Chest CT Scans with COVID-19 Related Findings dataset (URL: https://mosmed.ai/datasets/covid19_1110), which is freely available for the purpose of research. The dataset contains CT lung scans of 1110 patients with or without COVID-19 infections. The images were acquired between March 1st, 2020 and April 25th, 2020, and are maintained by medical hospitals in Moscow, Russia. The dataset also contains 50 labeled images of GGOs and their consolidations. The CT images in the dataset come from 42% male, 56% female, and 2% other categories of patients. We also evaluated the performance of our trained model on the datasets mentioned in the literature [22–24].

### §3.1 Segmentation of GGOs in lungs

Currently, there are only a few *labeled* data sets of CT lung scans of COVID-19 patients. Therefore, supervised deep learning (DL) approaches such as Convolutional Neural Network (CNN) or Recurrent Neural Network (RNN) cannot handle the complexity of the raw 3D data for lungs and GGOs in CT scans. We have circumvented this problem by using an unsupervised DL approach to segment GGOs in unlabeled MosMedData. For algorithmic processing, we extract 2D slices from 3D data while retaining the ability to reconstruct the 3D volume whenever required. Our 3D lung scans are available in Neuroimaging Informatics Technology Initiative (NIfTI) format. We applied the correct windowing filters to retain the regions of interest (ROIs), i.e., whole lung regions and GGOs. We use image processing operations to segment CT images irrespective of how they are coded for healthy and diseased cases. To do this effectively, we first remove skeletal and other background objects from an image slice using a simple thresholding algorithm (see S1 File) with the minimum and maximum values of the image intensity set between 117 and 255, respectively. Next, the resulting image is fed to a marker function which separates the lung from other background objects using morphological operations such as dilation and erosion (see S1 File). The dilation operation makes the lung regions more prominent by filling any small gaps, and the erosion operation removes any unnecessary objects. Lastly, we apply a structuring element operation to fill any gaps missed by the dilation operation (see S1 File). The output is a binary image with '1' representing the lung regions and '0' representing the background.

Next, we use unsupervised approaches to segment lung and GGOs. Fig 7 (see S1 File) (top panel) shows healthy and COVID-19 infected lung images. The diffused GGO region is inside

**Table 1. Summary of existing studies.**

| Objectives | Data | Method | Results | Reference(s) |
|---|---|---|---|---|
| GGOs segmentation or detection | Non-COVID-19 CT | Level Set | Segmentation accuracy not reported | [13] |
| | | k-Nearest Neighbor | 3.70% Mean error | [14] |
| | COVID-19 CT | Manual approach | ~36% Error rate | [15] |
| | | | GGOs detected manually | [16] |
| | | MSD-Net | 74.22% Dice similarity (DS) | [17] |
| | | DASC-Net | 76.33% DS | [18] |
| Risk Prediction | COVID-19 Clinical data | Neural Network | 77.6% accuracy | [19] |
| | | | 97% Area under the curve | [20] |
| Point cloud and PointNet/PointNet++ | Non-COVID-19 data | GRNet | 59.1% mAP@0.25 and 39.1% mAP @0.5 on ScanNetV2 dataset | [21] |

the red square. An expanded view of the GGO region is shown in Fig 7 (see S1 File) (bottom panel). The binary masks generated for lung segmentation misses GGO regions. Hence, we applied a convex hull algorithm (see S1 File) to preserve the actual anatomical structure of the lungs. The convex hull algorithm detects convex points around the outer surface of both lungs. The inner surface of the lungs was left untouched.

Fig 1 shows the workflow for lung segmentation. The original image (Fig 1A) and its binary mask leave an incomplete structure of the lung because the pixel intensities of GGO and soft tissue regions are very similar to each other and hard to delineate. Moreover, some parts of the lung are missing after masking. We apply the convex hull algorithm (see S1 File) to address these problems caused by masking. The convex hull detection results are shown in Fig 1C. The red dots around the edges of both lungs represent convex points. Next, we computed pairwise Euclidean distances and drew a green line if any pairwise distance was less than 80 pixels (this value may vary with the dataset). We connected the missing GGO regions from the segmented lung images in such a way that the original lung structure was restored (Fig 1D and 1E). Finally, we converted the binary mask into an RGB mask using bitwise operation between binary masks and original images.

To segment GGOs, we segregated individual clusters of RGB lung masks using the *K*-means algorithm. The GGOs were assigned to the cluster 1, lungs without GGOs to cluster 2, and the background corresponding to pixel values zeros to cluster 3. We applied connected component algorithms to remove unwanted small objects from the final results. The GGO detection results marked with red color are shown in the supplementary material, see Fig 8 in S1 File.

## §4 Results

We have used AI approaches and tensor analysis to characterize GGOs segmented from images in MosMedData. This section describes the use of PointNet++ for 3D visualization, classification, segmentation, and spatial distribution of GGOs. Next, we discuss our model for abnormality assessment and predictions based on the PointNet++ analysis. Lastly, we discuss how GGO shapes and sizes are quantified with the Minkowski tensor approach.

### §4.1 Point cloud and PointNet++ for pattern analysis of GGO distribution

A point cloud is a set of geometric data points in which each point represents a 3D centroid in Cartesian coordinates [25–27]. The PointNet and PointNet++, developed by the Stanford group

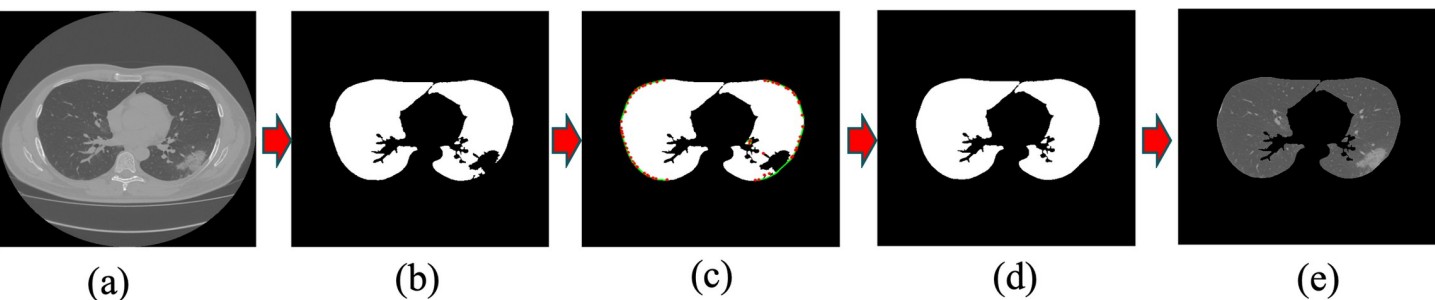

(a)                 (b)                 (c)                 (d)                 (e)

**Fig 1.** Shows the workflow of lung segmentation: In (a) we show an original image of lung CT scan, and (b) shows a binary mask of the original image. Since the pixel intensities of GGO and bone regions are similar to each other, the binary mask shows an incomplete structure of the lung. Some parts of the lungs are missing. To restore the missing parts, we applied the convex hull algorithm to the binary mask. The convex hull algorithm detects convex points [red dots in Fig 1C], and we compute pairwise Euclidean distances to connect nearest-neighbor points by a green line if the distance between them is less than 80 pixels [Fig 1C]. Fig 1D and 1E show the restored mask and the corresponding RGB image, respectively.

[28–30], use point cloud data as input to segment and classify 3D objects in a much more efficient way with less memory overhead than pixel or voxel-based methods such as CNN.

PointNet or PointNet++ has rarely been used for medical image analysis. To the best of our knowledge, we are the first group to use the PointNet++ architecture for 3D visualization, segmentation, classification, and pattern analysis of GGOs. The main challenge in applying PointNet++ to CT scans of COVID-19 patients is the data conversion from images to the point cloud representation and the training of PointNet++. These challenges are addressed by developing ML algorithms, which convert segmented lung images (described in section 3.1) to Hounsfield units (HU) and then to point cloud. They respect the permutation invariance of the input points and are more robust with respect to input perturbations.

We used the segmented lungs and GGOs as the main data points. We selected 16 slices from sets of 25 or more slices and kept those representing the full lung and removed others containing just lungs or soft tissue parts. Incidentally, from the algorithmic point of view, there are no restrictions on the number of slices one can include in the point cloud representation. We computed centroid points using the HU data values. Fig 2 shows the workflow diagram of the point cloud and PointNet++ for pattern analysis of GGO distribution.

The sixteen slices we chose generated close to 800,000 geometric data points. We reduced the number of data points by down-sampling and grouping centroid points to represent the local neighborhood around each point. In this way, we reduced the number of points in point clouds to 2048 without affecting the properties of point clouds or the actual structure of lung images. Next, we trained the PointNet++. As shown in Fig 2, the PointNet++ had a multiple set of abstraction layers forming a hierarchical feature learning architecture [28, 29], which merged local and global features to get the score of individual data points and used two fully connected layers to extract global features for classification.

The training and testing curves for PointNet++ are shown in Fig 3 and the performance of PointNet++ in Table 2. The evaluation accuracy of PointNet++ is 97.64% and its evaluation intersection over union is 92.20%. The results of PointNet++ are displayed in Fig 4, where cyan color bubbles represent lung and green color bubbles represent GGOs. These 3D results are constructed with sixteen slices. The 3D renditions can be rotated in any direction to view the distribution pattern of GGOs over the entire stack (see the movie in the supplementary material).

## §4.2 Automated abnormality classification

Abnormality score classification is a vital part of self-interpretable AI and anomaly detection [31]. We computed the abnormality of individual regions of the segmented lung and GGOs (described in section 3.1) using a CNN network presented in Fig 5. This network is the same as VGG-16 except for the first four convolution layers which were reduced to two. These architectural changes achieved better performance. The first four convolution layers of the VGG-16 network accept images of size 224 x 224 pixels and 112 x 112pixels. Our input data size is 512 x 512pixels. To accommodate our images into VGG-16, we had to resize the image. But resizing large images into smaller sizes at the initial stage has a chance of information loss. Hence, we have replaced the four convolution layers with two layers. Accordingly, we have changed the kernel size and max-pooling operations. Moreover, in the VGG-16 network, the final layer is the fully connected layer. In our case, the final layer is the Cox layer, which computes abnormality scores. If we use exactly the VGG-16 network, we will not be able to compute abnormality scores. The CNN network is comprised of multiple convolutions, max-pooling, and dense layers at the end of the network. Rectified linear units were used after each convolution layer, and an adaptive learning technique with log-likelihood loss functions was used for the training

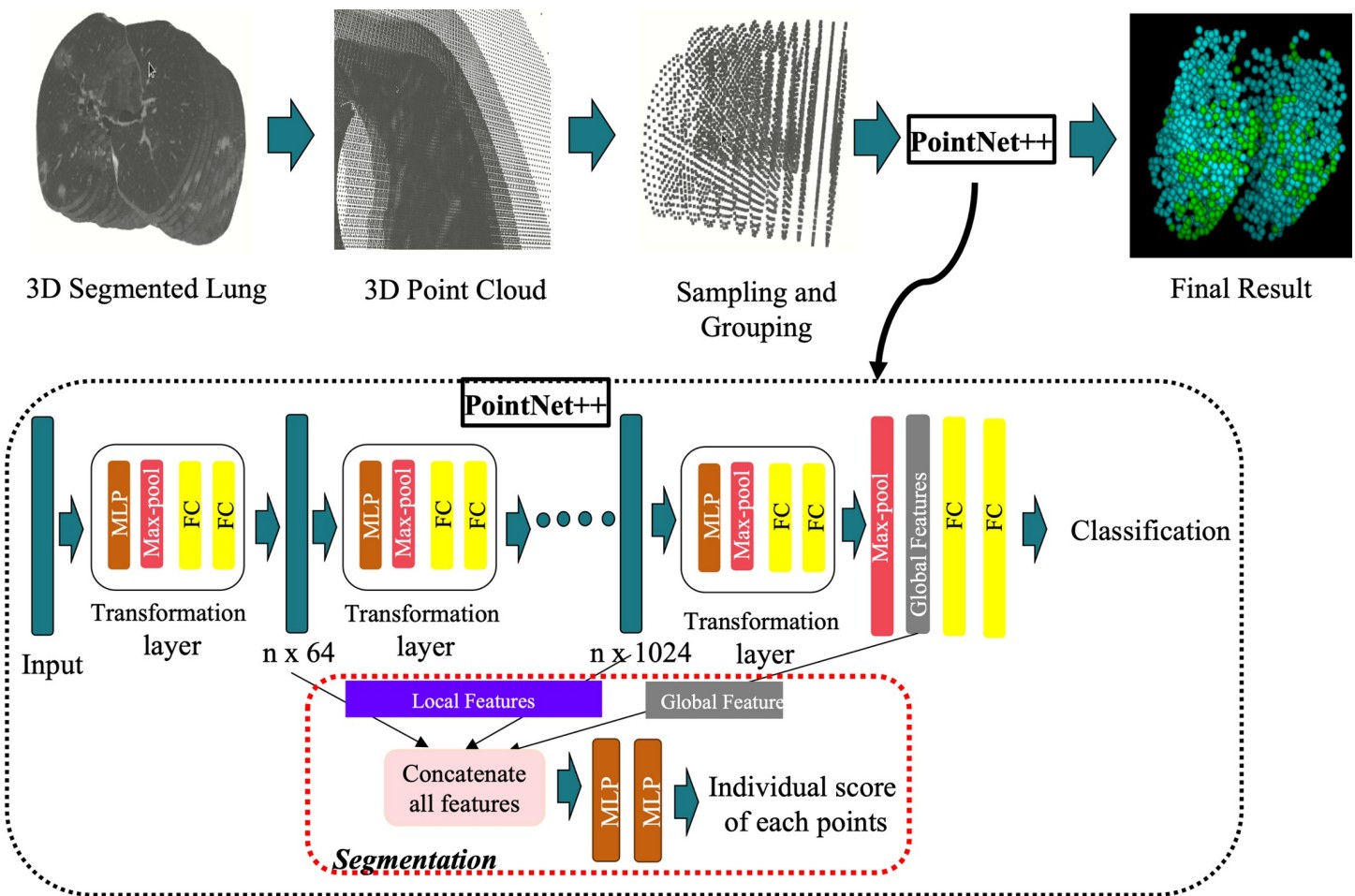

**Fig 2. Shows the workflow diagram of the point cloud and PointNet++ for pattern analysis of GGO distribution.** The PointNet++ architecture used in this study is the same as the original architecture in [28 and 29]. The data preparation pipeline is new, which we developed for generating point clouds from 3D lung CT scans. We stacked all segmented lung images to construct a 3D image and used sampling and grouping to select 2048 points for point clouds. We fed those points to the PointNet++ architecture, which was constructed with a multiple set of abstraction layers. Each set abstraction layer consisted of a sampling layer, grouping layer, and PointNet [28]. Multiple set abstraction layers formed a hierarchical feature learning architecture, which was divided into two parts: Segmentation and classification. For data segmentation, we merged local and global features to get the score of individual data points. For classification, global features were fed to two fully connected (FC) layers. MLP—Multilayer Perceptron.

of the network. The dropout was set at 0.5, the initial learning rate to 0.001, and the model was trained up to 60 epochs.

The output of the last dense layer was fed to the Cox proportional hazard model (see S1 File) [32, 33]. This regression model is generally used for computing abnormality scores and survival analysis. The model computed scores of individual regions of segmented lung and GGOs. Based on the values of the Cox model, low-abnormality and high-abnormality regions were determined. The regions which belong to a healthy lung or background were considered low abnormality regions. The regions containing GGOs or some other malignant signatures like a tumor nodule were deemed high-abnormality regions. Fig 5A shows the CNN architecture for automated abnormality classification, and the workflow diagram of the methodology is shown in Fig 5B. The original image was fed to a CNN and then the Cox model was used to construct the abnormality heatmap. Low and high abnormality regions are colored black and red, respectively. Fig 9 in S1 File shows regions of abnormality in many CT images.

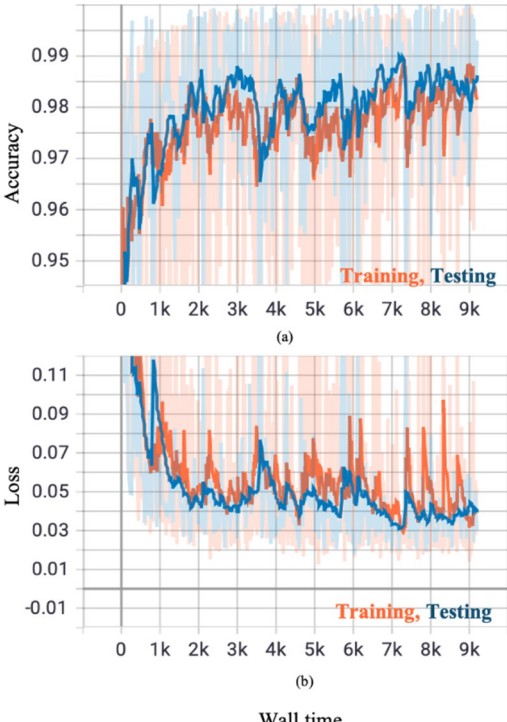

**Fig 3. Shows the PointNet++ training and testing curves.** Training and testing accuracies increase with increasing wall time and concomitantly the loss decreases with increasing wall time. This is an ideal behavior of a properly trained model.

## §4.3 Shape analysis of GGOs

GGOs are highly anisotropic objects. Scalar quantities such as volume and surface area cannot capture anisotropic features of GGOs. Therefore, we use Minkowski tensors to characterize the shapes of GGOs [34, 35]. Minkowski tensors are based on the mathematical foundations of integral and stochastic geometry. A simple example of shape representation by a tensor is the moment of inertia of a rotating body about an axis. Generalizing the concept of volume, surface area and local curvatures, Minkowski tensors quantify the morphologies of anisotropic objects in terms of shape indices. Minkowski functionals are rank-2 tensors and have four independent real-valued components in $d = 2$ spatial dimensions and six in $d = 3$ dimensions. For a three-dimensional object $K$ with the surface $\partial K$, the six tensor components are given by,

$$W_0^{2,0}(K) = \int \boldsymbol{r} \otimes \boldsymbol{r}dV \; ; \; W_1^{2,0}(K) = \frac{1}{3}\int \boldsymbol{r} \otimes \boldsymbol{r}dA; \; W_2^{2,0}(K) = \frac{1}{3}\int \boldsymbol{H}(\boldsymbol{r})\boldsymbol{r} \otimes \boldsymbol{r}dA$$

$$W_3^{2,0}(K) = \frac{1}{3}\int \boldsymbol{G}(\boldsymbol{r})\boldsymbol{r} \otimes \boldsymbol{r}dA \; ; \; W_1^{0,2}(K) = \frac{1}{3}\int \boldsymbol{n} \otimes \boldsymbol{n}dA \; ; \; W_2^{0,2}(K) = \frac{1}{3}\int \boldsymbol{H}(\boldsymbol{r})\boldsymbol{n} \otimes \boldsymbol{n}dA$$

**Table 2. Performance results of PointNet++.**

| | |
|---|---|
| Evaluation Mean Loss | 0.0613 ± 0.0477 |
| Evaluation Accuracy | 0.9764 ± 0.0210 |
| Evaluation Average Class Accuracy | 0.9517 ± 0.04751 |
| Mean Intersection over Union (MIoU) | 0.9220 ± 0.05630 |

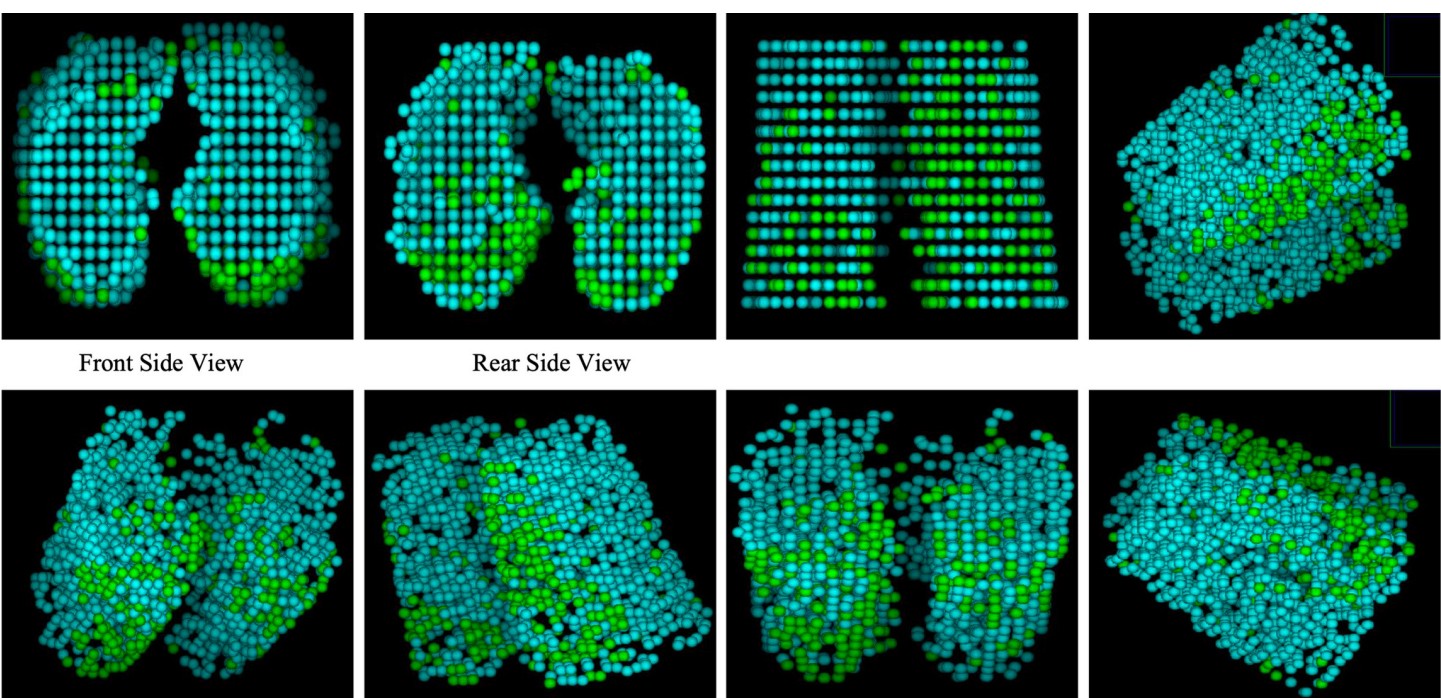

**Fig 4. Final results after training the PointNet++ using point cloud data are shown here.** The 3D images show different regions from different angles. Cyan color points/bubbles represent non-GGO regions and green color points/bubbles represent GGO regions.

where $H(\boldsymbol{r}) = (\kappa_1 + \kappa_2)/2$ and $G(\boldsymbol{r}) = (\kappa_1 \kappa_2)$ are the mean and Gaussian curvatures, $\boldsymbol{n}(\boldsymbol{r})$ is the normal vector of $\partial K$ at $\boldsymbol{r}$, and $\otimes$ denotes the tensor product $(a \otimes a)_{ij} = a_i a_j$.

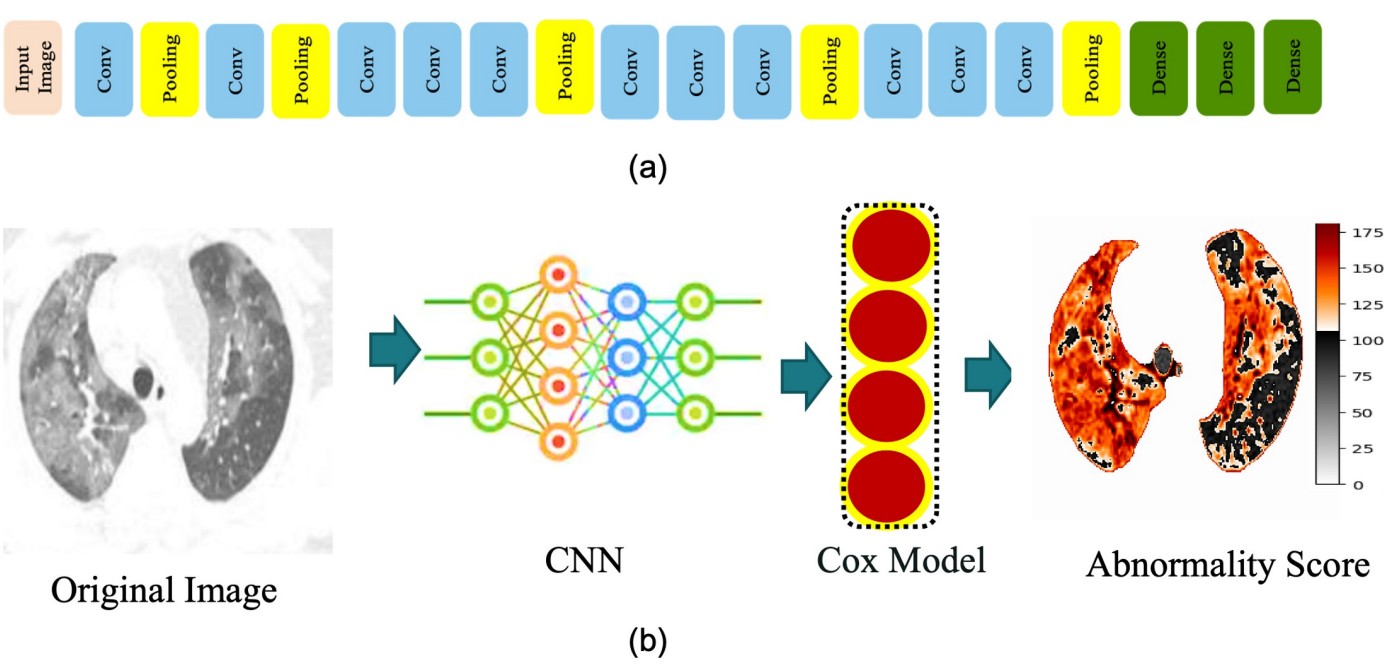

**Fig 5.** (a) The architecture of the CNN for abnormality prediction. We used the segmented lungs and GGOs as input images (described in section 3.1). (b) Workflow diagram for abnormality prediction. We used the CNN architecture mentioned in Fig 5A. The output of the CNN network was fed to the Cox model to compute abnormality. The output of abnormality scores is represented by a heatmap. Orange to red color denotes high-abnormality regions.

Geometric methods have been developed for biological shape analysis, and shape-based methods have been used for diagnosis of lung nodules [36–38]. Minkowski tensor-based shape analysis is a new approach to analyze shapes of GGOs. They have been used in general theory of relativity and materials science. Highly efficient algorithms have been developed to compute Minkowski tensors in linear time (URL: www.theorie1.physik.uni-erlangen.de), and they have been used to characterize a variety of systems: (a) surface force microscopy images of co-polymer films and X-ray tomography images of open-cell liquid foams; (b) models of liquid foams and granular materials consisting of spherical beads; and (c) defect structures in molecular dynamics simulations of metals.

We characterize anisotropies in GGOs in terms of the eigenvalues of Minkowski tensors. The degree of anisotropy is expressed as the ratios of the minima to maxima of eigenvalues:

$$\beta_v^{r,s} := \frac{|\mu_{min}|}{|\mu_{max}|} \in [0,1]$$

where $\mu_{\min}$ is the minimum and $\mu_{\max}$ is the maximum eigenvalue of the tensor $W_v^{r,s}$. The absolute value is introduced because the tensor $W_0^{2,2}$ can have negative eigenvalues for non-convex objects. For isotropic bodies, $\beta_v^{r,s} = 1$, and deviations from unity reflect the degree of anisotropy.

We analyzed the shapes of individual GGOs using Minkowski tensors and their eigenvalues and computed areas, perimeters, area fractions, isoperimetric ratios, anisotropy indices, and different quadrupole moments ($q_2$, $q_3$, $q_4$, $q_5$, and $q_6$) of GGOs. The results are shown in Fig 6. The probability distribution in Fig 6A indicates a peak around -600 HU and a tail extending to -150 HU for GGOs in the dataset. A majority of GGOs in the dataset lie in the tail region corresponding to high-abnormality scores in the heatmap of Fig 5. The average area and perimeters of GGOs in image slices are 3821 and 578, respectively. The $\beta_1$ index is around 0.85, which

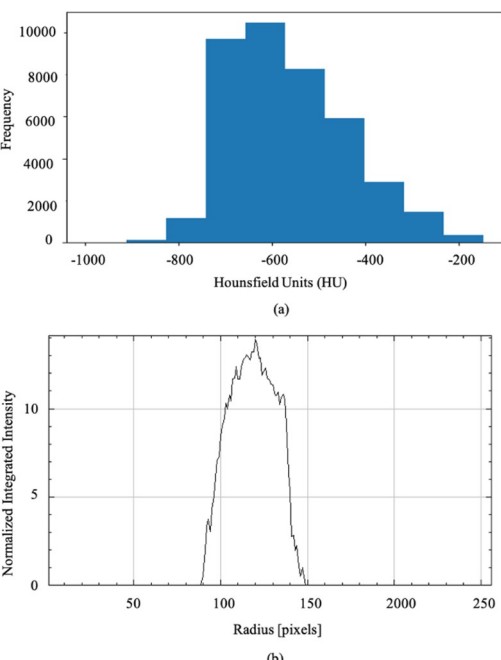

**Fig 6. GGO shape analysis of COVID-19 data using Minkowski tensors.** (a) Probability distribution shows that GGOs for all the images lie between -800 and -150 HU with peaks in the distribution centered around -600 HU. (b) Radial distribution function versus distance shows a peak around 125 pixels.

indicates 15% asphericity for GGOs. The isoperimetric ratio is 0.144. Among the moments, $q_3$ and $q_6$ are the largest and the dipole moment $q_2$ is the second largest. We also computed the radial distribution function, $g(r)$, and pixel radius and normalized integrated intensity (Fig 6B). The radial distribution function lies between 90 and 150 pixels, peaking around 120 which indicates that pixels in GGOs are correlated over long distances.

## §5 Conclusion

We have demonstrated the efficacy of an end-to-end unsupervised deep learning approach to modeling GGOs segmented from CT images of the lungs in COVID-19 patients. The segmentation and classification approach, PointNet++, can accurately detect GGOs in 3D CT scans. Instead of modeling all the pixels of a GGO, we use point cloud feature learning in which a small fraction of pixels is sampled randomly and the 3D shape of a GGO is faithfully reconstructed. In addition to its efficiency, our approach has the benefit of representing the local neighborhood of a point using features that can be used in downstream ML algorithms. The accuracy of PointNet++ in detecting GGOs is 98%, the average class accuracy is 95%, and the intersection over union is 92%. To the best of our knowledge, no other group has demonstrated the segmentation or quantified the morphologies of GGOs from 3D CT scans of lungs.

We have used a mathematical approach based on Minkowski tensors to quantify the size and shape distributions of GGOs. On average, the shapes of GGOs in the COVID-19 datasets deviate from sphericity by 15% and anisotropies in GGOs are dominated by dipole and hexapole components. These anisotropies may help to quantitatively delineate GGOs of COVID-19 from other lung diseases. Furthermore, the 3D modeling of GGO in the lung enhances identification of classic disease patterns in COVID-19.

The PointNet++ and the Minkowski tensor based morphological approach together with abnormality analysis will provide radiologists and clinicians with a valuable set of tools when interpreting CT lung scans of COVID-19 patients. Implementation would be particularly useful in countries severely devastated by COVID-19 such as India, where the number of cases has outstripped available resources creating delays or even breakdowns in patient care. This AI-driven approach synthesizes both the unique GGO distribution pattern and severity of the disease to allow for more efficient diagnosis, triaging and conservation of limited resources.

### Software and hardware

Our model was trained using TensorFlow (r2.3) on NVIDIA DGX-1 servers equipped with eight NVIDIA V100 GPUs. All the algorithms were developed using Python.

### Codes

The relevant codes will be made available for the public upon acceptance of this manuscript.

### Supporting information

**S1 File. This is the supporting material file.**
(ZIP)

**S2 File. This file contains some additional results.**
(PPTX)

### Author Contributions

**Conceptualization:** Monjoy Saha, Ashish Sharma, Rajiv K. Kalia.

**Data curation:** Monjoy Saha, Rajiv K. Kalia.

**Formal analysis:** Monjoy Saha.

**Funding acquisition:** Ashish Sharma, T. K. Satish Kumar, Rajiv K. Kalia.

**Investigation:** Monjoy Saha, Ashish Sharma, Rajiv K. Kalia.

**Methodology:** Monjoy Saha.

**Project administration:** Ashish Sharma, T. K. Satish Kumar, Rajiv K. Kalia.

**Resources:** Monjoy Saha.

**Software:** Monjoy Saha.

**Supervision:** Ashish Sharma, T. K. Satish Kumar.

**Validation:** Monjoy Saha, Sagar B. Amin.

**Visualization:** Monjoy Saha, Sagar B. Amin, Ashish Sharma.

**Writing – original draft:** Monjoy Saha, Rajiv K. Kalia.

**Writing – review & editing:** Monjoy Saha, Sagar B. Amin, Ashish Sharma, T. K. Satish Kumar, Rajiv K. Kalia.

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
