## [Decision Letter · Decision Letter 0]

6 Jan 2022

PONE-D-21-35862AI-DRIVEN QUANTIFICATION OF GROUND GLASS OPACITIES IN LUNGS OF COVID-19 PATIENTS USING 3D COMPUTED TOMOGRAPHY IMAGINGPLOS ONE

Dear Dr. Saha,

Thank you for submitting your manuscript to PLOS ONE. After careful consideration, we feel that it has merit but does not fully meet PLOS ONE’s publication criteria as it currently stands. Therefore, we invite you to submit a revised version of the manuscript that addresses the points raised during the review process. Your manuscript has been evaluated by two experts in the field and their comments are attached here below for you reference. You will see that, in particular, the reviewers requested additional details about the methods and datasets used in this study.

We look forward to receiving your revised manuscript.

Kind regards,

Francesco Bianconi, Ph.D.

Academic Editor

PLOS ONE

Journal Requirements:

“Ashish Sharma would like to acknowledge support from the National Cancer Institute U24CA215109. The content of this publication does not necessarily reflect the views or policies of the Department of Health and Human Services, nor does mention of trade names, commercial products, or organizations imply endorsement by the U.S. Government. T. K. Satish Kumar and Rajiv K. Kalia (RKK) would like to acknowledge the support of Zumberge Research and Innovation Fund at the University of Southern California. RKK would like to thank Sarah Kalia for helpful discussions.”

“Ashish Sharma would like to acknowledge support from the National Cancer Institute U24CA215109. The content of this publication does not necessarily reflect the views or policies of the Department of Health and Human Services, nor does mention of trade names, commercial products, or organizations imply endorsement by the U.S. Government. T. K. Satish Kumar and Rajiv K. Kalia (RKK) would like to acknowledge the support of Zumberge Research and Innovation Fund at the University of Southern California. RKK would like to thank Sarah Kalia for helpful discussion”

“Ashish Sharma would like to acknowledge support from the National Cancer Institute U24CA215109. The content of this publication does not necessarily reflect the views or policies of the Department of Health and Human Services, nor does mention of trade names, commercial products, or organizations imply endorsement by the U.S. Government. T. K. Satish Kumar and Rajiv K. Kalia (RKK) would like to acknowledge the support of Zumberge Research and Innovation Fund at the University of Southern California. RKK would like to thank Sarah Kalia for helpful discussion”

Reviewers' comments:

Reviewer's Responses to Questions

**Comments to the Author**

1. Is the manuscript technically sound, and do the data support the conclusions?

Reviewer #1: Yes

Reviewer #2: Yes

2. Has the statistical analysis been performed appropriately and rigorously? 

Reviewer #1: Yes

Reviewer #2: Yes

3. Have the authors made all data underlying the findings in their manuscript fully available?

Reviewer #1: Yes

Reviewer #2: Yes

4. Is the manuscript presented in an intelligible fashion and written in standard English?

Reviewer #1: Yes

Reviewer #2: Yes

5. Review Comments to the Author

Reviewer #1: The proposed method is good. Manuscripts have been written with the standards of Scientific Writing Publications. Manuscript need equip with a recapitulation of test results for all tested data. References from old journal articles changed to the latest issue.

Reviewer #2: the author proposed a manuscript which has a title "AI-DRIVEN QUANTIFICATION OF GROUND GLASS OPACITIES IN LUNGS OF COVID-19 PATIENTS USING 3D COMPUTED TOMOGRAPHY IMAGING". This research come up with a hot issues nowdays. The manuscript is explained structurely.

But need some explaination for some parts as follow

1. in line 252, author mentions that the network is the same as VGG-16 but author reduce the first four convolutiion layers into two. It was done to achieve best performance. how author proof this one ?

2. author mention that, the MosMedData is used in this research. what is the image size ? because if author modified model from VGG-16 which has its on pixel size ? what kind of preprocessing was done to the dataset ?

6. PLOS authors have the option to publish the peer review history of their article (what does this mean?). If published, this will include your full peer review and any attached files.

Reviewer #1: No

Reviewer #2: **Yes: **Ir. Rahmat Hidayat

---

## [Author Response · Author response to Decision Letter 0]

25 Jan 2022

Reviewer #1: The proposed method is good. Manuscripts have been written with the standards of Scientific Writing Publications. Manuscript need equip with a recapitulation of test results for all tested data. References from old journal articles changed to the latest issue.

Response: We appreciate the reviewer's insightful comments and ideas. We have uploaded the step-by-step results of our analysis to Google Drive. The below Google Drive link is available in the Supplementary Material.

https://drive.google.com/drive/folders/1q8U5JesS6DgStYXohaGisl2y39rSS4RC?usp=sharing

For the original dataset, please check the Materials and methods of the manuscript. We are unable to upload original data as redistribution of original data is not allowed. We have also added one "Additional_results.pptx" file as a Supplementary Material, which contains animated videos and additional results. 

As suggested by the reviewer, we also have updated the below sentence in our revised manuscript and accordingly updated the references. All the changes have been highlighted using yellow color.

Main Manuscript [page 3]

It has infected over 342 million people and killed over 5.5 million people at the time of writing this paper (source: https://coronavirus.jhu.edu/map.html).

References of Supplementary Material:

[1] Stare J, Maucort-Boulch D. Odds ratio, hazard ratio and relative risk. Metodoloski zvezki. 2016;13(1):59.

Additional_results.pptx. This file contains some additional results

Reviewer #2: 

Comment: The author proposed a manuscript which has a title "AI-DRIVEN QUANTIFICATION OF GROUND GLASS OPACITIES IN LUNGS OF COVID-19 PATIENTS USING 3D COMPUTED TOMOGRAPHY IMAGING". This research come up with a hot issue nowadays. The manuscript is explained structurally.

Response: The authors are thankful to the reviewer for their constructive and inspirational comments. We have addressed all the comments raised by the reviewer. Changes have been highlighted in the revised manuscript using yellow color.

Q1. in line 252, author mentions that the network is the same as VGG-16 but author reduce the first four convolution layers into two. It was done to achieve best performance. how author proof this one?

Response: Thank you so much for pointing out this question. Please find our response below. The below sentences have been updated in the revised manuscript. The changes have been highlighted using yellow color [page 13].

The first four convolution layers of the VGG-16 network accept images of size 224x224 pixels and 112x112pixels. Our input data size is 512x512pixels. To accommodate our images into VGG-16, we had to resize the image. But resizing large images into smaller sizes at the initial stage has a chance of information loss. Hence, we have replaced the four convolution layers with two layers. Accordingly, we have changed the kernel size and max-pooling operations. Moreover, in the VGG-16 network, the final layer is the fully connected layer. In our case, the final layer is the Cox layer, which computes abnormality scores. If we use exactly the VGG-16 network, we will not be able to compute abnormality scores. 

Q2. author mention that, the MosMedData is used in this research. what is the image size? because if author modified model from VGG-16 which has its on pixel size? what kind of preprocessing was done to the dataset?

Response: Thank you so much for asking this question. The image size of MosMed data is 512 x 512 pixels. We have modified the first four convolution layers of the VGG-16 network to accommodate the 512 x 512 pixels images. Accordingly, we have changed kernel size and max-pooling operations. We have updated this information in the revised manuscript.

---

## [Editor Report · Decision Letter 1]

31 Jan 2022

AI-DRIVEN QUANTIFICATION OF GROUND GLASS OPACITIES IN LUNGS OF COVID-19 PATIENTS USING 3D COMPUTED TOMOGRAPHY IMAGING

PONE-D-21-35862R1

Dear Dr. Saha,

We are pleased to inform you that your manuscript has been judged scientifically suitable for publication and will be formally accepted once it meets all outstanding technical requirements.

Within one week, you will receive an e-mail detailing the required amendments. When these have been addressed, you will receive a formal acceptance letter and your manuscript will be scheduled for publication.

If your institution or institutions have a press office, please notify them about your upcoming paper to help maximize its impact. If they will be preparing press materials, please inform our press team as soon as possible -- no later than 48 hours after receiving the formal acceptance. Your manuscript will remain under strict press embargo until 2 pm Eastern Time on the date of publication. For more information, please contact onepress@plos.org.

Kind regards,

Francesco Bianconi, Ph.D.

Academic Editor

PLOS ONE

---

## [Editor Report · Acceptance letter]

7 Feb 2022

PONE-D-21-35862R1 

AI-driven Quantification of Ground Glass Opacities in lungs of covid-19 patients using 3D Computed Tomography Imaging 

Dear Dr. Saha:

I'm pleased to inform you that your manuscript has been deemed suitable for publication in PLOS ONE. Congratulations! Your manuscript is now with our production department. 

Kind regards, 

on behalf of

Prof. Francesco Bianconi 

Academic Editor

PLOS ONE